# Dual-Head Knowledge Distillation: Enhancing Logits Utilization with an Auxiliary Head

## Abstract

Traditional knowledge distillation focuses on aligning the student's predicted probabilities with both ground-truth labels and the teacher's predicted probabilities. However, the transition to predicted probabilities from logits would obscure certain indispensable information. To address this issue, it is intuitive to additionally introduce a logit-level loss function as a supplement to the widely used probability-level loss function, for exploiting the latent information of logits. Unfortunately, we empirically find that the amalgamation of the newly introduced logit-level loss and the previous probability-level loss will lead to performance degeneration, even trailing behind the performance of employing either loss in isolation. We attribute this phenomenon to the collapse of the classification head, which is verified by our theoretical analysis based on the *neural collapse* theory. Specifically, the gradients of the two loss functions exhibit contradictions in the linear classifier yet display no such conflict within the backbone. Drawing from the theoretical analysis, we propose a novel method called *dual-head knowledge distillation*, which partitions the linear classifier into two classification heads responsible for different losses, thereby preserving the beneficial effects of both losses on the backbone while eliminating adverse influences on the classification head. Extensive experiments validate that our method can effectively exploit the information inside the logits and achieve superior performance against state-of-the-art counterparts.

## 1 Introduction

Despite the remarkable success of deep neural networks (DNNs) in various fields, it is a significant challenge to deploy these large models in lightweight terminals (e.g., mobile phones), particularly under the constraint of computational resources or the requirement of short inference time. To mitigate this problem, knowledge distillation (KD) (Hinton et al., 2015) is widely investigated, which aims to improve the performance of a small network (*a.k.a.* the "student") by leveraging the expansive knowledge of a large network (*a.k.a.* the "teacher") to guide the training of the student network.

Traditional KD techniques focus on minimizing the disparity in the predicted probabilities between the teacher and the student, which are typically the outputs of the softmax function. Nevertheless, the transformation from logits to predictive probabilities via the softmax function may lose some underlying information. As shown in Figure 1(a), considering a 3-class classification problem, even if the teacher model outputs two different logit vectors $[2, 3, 4]$ and $[-2, -1, 0]$, the softmax function renders the same probability vector $[0.09, 0.24, 0.67]$. However, different logit vectors may carry different underlying information that would be further exploited by the student, which could be lost due to the transformation process carried out by the softmax function.

In order to properly leverage the information inside the logits, we introduce a logit-level KD loss. Specifically, we use the sigmoid function to formalize the pre-softmax output for each class into a range of $[0, 1]$ and deploy the Kullback-Leibler (KL) divergence to perform the binary classification of each class. By aligning the pre-softmax output of each class separately, this newly introduced loss (denoted by *BinaryKL*) can adequately exploit the information inside the logits. However, it is interesting to show that combining the logit-level BinaryKL loss with the probability-level cross-entropy (CE) loss results in poor performance of the student model, which is even worse than the performance of employing either loss separately. As illustrated in Figure 1(b), employing either

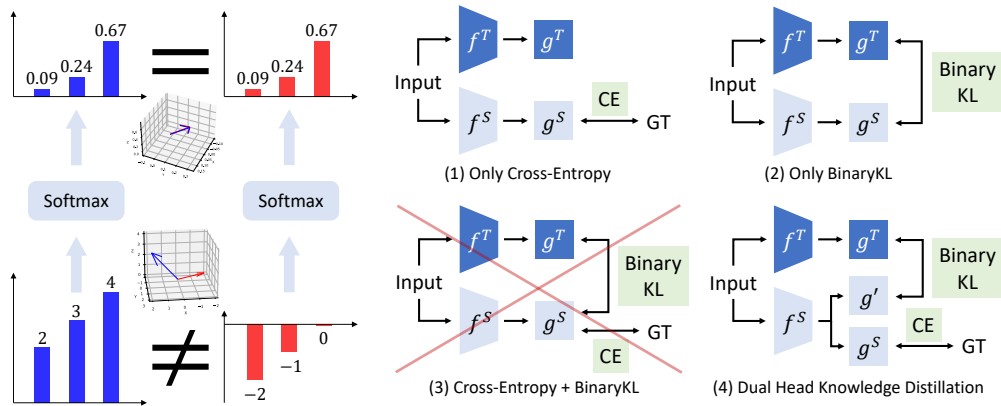

(a) Information loss through softmax          (b) An illustration of four settings we evaluate

Figure 1: **The reason for introducing the BinaryKL loss and the incompatibility of the CE loss and the BinaryKL loss.** Figure 1(a) shows that two different vectors may become the same through the softmax function, which means some information would be lost during the transformation process carried out by the softmax function. Figure 1(b) shows four settings we evaluate: (1) only cross-entropy (CE) loss; (2) only the BinaryKL loss; (3) CE + BinaryKL; (4) Our proposed Dual-Head Knowledge Distillation (DHKD). The red cross over the third setting means that the student model trained under this setting will collapse. The performance sorting of four settings is (4) > (1) > (2) ≫ (3), as shown in Figure 2.

loss separately as the training loss can induce a high-performance student model, but the model's performance will be largely degraded when we combine them together during the training process.

To identify the root cause of this abnormal phenomenon, inspired by the recent research on *neural collapse* (Yang et al., 2022), we analyze the gradients of the model when using both losses. Specifically, by dividing the gradients of the model into two parts, we find that the gradients of the two loss functions contradict each other regarding the linear classifier head but display no such conflict regarding the backbone. As a consequence, while the BinaryKL loss can facilitate the backbone in learning from the teacher model more precisely, it prevents the linear classifier head from converging to a simplex equiangular tight frame (see a detailed definition in Definition 1), which is the ideal status that a well-trained linear classifier should converge to. Therefore, the combination of these two loss functions would cause the linear classifier to collapse, thereby degrading the performance of the student model. Based on our theoretical analysis that a single linear classifier is faced with the gradient contradiction when being trained by both losses simultaneously, we propose Dual-Head Knowledge Distillation (DHKD), which introduces an auxiliary classifier head apart from the original linear classifier to effectively circumvent the collapse of the linear classifier while retaining the positive effects of the BinaryKL loss on the backbone. Extensive experiments show the advantage of our proposed DHKD method.

Our main contributions can be summarized as follows:

- We disclose an interesting phenomenon in the knowledge distillation scenario: combining the probability-level CE loss and the logit-level BinaryKL loss would cause a performance drop of the student model, compared with using either loss separately.

- We provide theoretical analyses to explain the discordance between the BinaryKL loss and the CE loss. While the BinaryKL loss aids in cultivating a stronger backbone, it harms the performance of the linear classifier head.

- We propose a novel knowledge distillation method called Dual-Head Knowledge Distillation (DHKD). Apart from the linear classifier trained with the CE loss, DHKD specially introduces an auxiliary classifier trained with the BinaryKL loss.

Extensive experiments demonstrate the effectiveness of our proposed method.

## 2 RELATED WORK

### 2.1 KNOWLEDGE DISTILLATION

Knowledge distillation (KD) (Hinton et al., 2015) aims to transfer knowledge from a large teacher network to a small student network. Existing works can be roughly divided into two groups: feature-based methods and logit-based methods.

Feature-based methods focus on distilling knowledge from intermediate feature layers. FitNet (Romero et al., 2015) is the first approach to distill knowledge from intermediate features by measuring the distance between feature maps. RKD (Park et al., 2019) utilizes the relations among instances to guide the training process of the student model. CRD (Tian et al., 2019) incorporates contrastive learning into knowledge distillation. OFD (Heo et al., 2019) contains a new distance function to distill significant information between the teacher and student using marginal ReLU. ReviewKD (Chen et al., 2021) proposes a review mechanism that uses multiple layers in the teacher to supervise one layer in the student. Other papers (Passalis & Tefas, 2018; Kim et al., 2018; Koratana et al., 2019; Li, 2022; Liu et al., 2023; Wang et al., 2023; Roy Miles & Deng, 2024; Miles & Mikolajczyk, 2024) enforce various criteria based on features. Most feature-based methods can attain superior performance, yet involving considerably high computational and storage costs.

Logit-based methods mainly concentrate on distilling knowledge from logits and softmax scores after logits. DML (Zhang et al., 2018b) introduces a mutual learning method to train both teachers and students simultaneously. DKD (Zhao et al., 2022) proposes a novel logit-based method to reformulate the classical KD loss into two parts and achieves state-of-the-art performance by adjusting weights for these two parts. DIST (Huang et al., 2022) relaxes the exact matching in previous KL divergence loss with a correlation-based loss and performs better when the discrepancy between the teacher and the student is large. TTM (Zheng & Yang, 2024) drops the temperature scaling on the student side, which causes an inherent Renyi entropy term as an extra regularization term in the loss function. Although logit-based methods require fewer computational and storage resources, they experience a performance gap compared with feature-based methods.

### 2.2 NEURAL COLLAPSE

Neural collapse is a phenomenon observed in the late stages of training a deep neural network, which is especially evident in the classification tasks (Papyan et al., 2020). As the training process approaches its optimum, the feature representations of data points belonging to the same class tend to converge to a single point, or at least become significantly more similar to each other in the feature space. At the same time, the class mean vectors tend to be equidistant and form a simplex equiangular tight frame (see a detailed definition in Definition 1).

Although the phenomenon is intuitive, its reason has not been entirely understood, which inspires several lines of theoretical work on it. Papyan et al. (2020) prove that if the features satisfy neural collapse, the optimal classifier vectors under the MSE loss will also converge to neural collapse. Some studies turn to a simplified model that only considers the last-layer features and the classifier as independent variables, and they prove that neural collapse emerges under the CE loss with proper constraints or regularization (Fang et al., 2021; Ji et al., 2022; Lu & Steinerberger, 2022; Weinan E, 2022; Zhu et al., 2021). Yang et al. (2022) also use such a simplified model and decompose the gradient into two parts to help build a better classifier for class-imbalanced learning, which is the primary technical reference for the theoretical analysis section of this article.

### 2.3 DECOUPLED HEADS

Using multiple linear classifiers is a common strategy in various fields. In object detection, to address the conflict between classification and regression tasks, the decoupled head for classification and localization is widely used in most one-stage and two-stage detectors (Ge et al., 2021; Wu et al., 2020; Lin et al., 2017). In multitask learning, it is common practice to jointly train various tasks through the shared backbone (Caruana, 1997; Zheng et al., 2022; Kendall et al., 2018). In long-tailed learning, the Bilateral-Branch Network takes care of both representation learning and classifier learning concurrently, where each branch performs its own duty separately (Zhou et al., 2020).

## 3 DUAL-HEAD KNOWLEDGE DISTILLATION

In this section, we firstly point out the incompatibility between the CE loss and the BinaryKL loss in Section 3.1. Then we analyze its theoretical foundation in Section 3.2. Finally, based on the theoretical analysis, we introduce our Dual-Head Knowledge Distillation (DHKD) method in Section 3.3.

### 3.1 INCOMPATIBILITY BETWEEN CE AND BINARYKL

Traditional KD methods only minimize the difference in the predicted probabilities (*i.e.*, the outputs of the softmax function) between the teacher and the student. However, some underlying information may be lost when converting logits to predicted probabilities using the softmax function. For example, in a 3-class classification problem in Figure 1(a), if the teacher model outputs logit vectors $[2, 3, 4]$ and $[-2, -1, 0]$ for different instances, after being processed by the softmax function, their predicted probabilities are both $[0.09, 0.24, 0.67]$. Such differences in logits may carry different hidden information that would be further utilized by the student but could be lost because of the transformation process carried out by the softmax function.

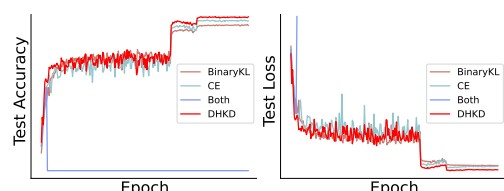

Figure 2: **The test accuracy and test loss curves of the student models during the training phase.** We set **resnet56** as the teacher and **resnet20** as the student on the CIFAR-100 dataset.

To fully exploit the information inside the logits, we introduce a logit-level KD loss. Previous studies in various fields have delved into some logit-level loss functions, such as the mean squared error and the mean absolute error (Kim et al., 2021). We explore a recently proposed method called BinaryKL (Yang et al., 2023), which uses the sigmoid function to formalize the pre-softmax output for each class into a range of $[0, 1]$ and deploys the Kullback-Leibler (KL) divergence as the binary classification of each class. By aligning the pre-softmax output of each class separately, the newly introduced loss can better exploit the information inside the logits.

Formally, with the logits $\boldsymbol{z}^{\mathcal{S}} \in \mathbb{R}^{B \times K}$ and $\boldsymbol{z}^{\mathcal{T}} \in \mathbb{R}^{B \times K}$ of the student model and the teacher model, where $B$ and $K$ denote batch size and the number of classes respectively, the BinaryKL loss can be formulated as follows:

$$\mathcal{L}_{\text{BinaryKL}} = \tau^2 \sum_{i=1}^{B} \sum_{k=1}^{K} \mathcal{KL}\left(\left[\sigma\left(z_{i,k}^{\mathcal{T}}/\tau\right), 1 - \sigma\left(z_{i,k}^{\mathcal{T}}/\tau\right)\right] \| \left[\sigma\left(z_{i,k}^{\mathcal{S}}/\tau\right), 1 - \sigma\left(z_{i,k}^{\mathcal{S}}/\tau\right)\right]\right), \quad (1)$$

where $\sigma(\cdot)$ is the sigmoid function, $[\cdot, \cdot]$ is an operator used to concatenate two scalars into a vector, and $\mathcal{KL}$ denotes the KL divergence $\mathcal{KL}(P\|Q) = \sum_{x \in \mathcal{X}} P(\boldsymbol{x}) \log\left(P(\boldsymbol{x})/Q(\boldsymbol{x})\right)$, where $P$ and $Q$ are two different probability distributions.

As shown in Figures 1 and 2, although the BinaryKL loss performs well alone, it is incompatible with the CE loss. The student model can achieve descent performance by training with either the CE loss or the BinaryKL loss separately, but the performance will be degraded severely when we combine them as follows:

$$\mathcal{L}_{\text{overall}} = \mathcal{L}_{\text{CE}} + \alpha \mathcal{L}_{\text{BinaryKL}}. \quad (2)$$

It seems that this problem can be addressed by decreasing the balancing parameter $\alpha$, reducing the learning rate, or using clipping the gradients, but our experiments show that these three solutions either do not work or weaken the effects of the BinaryKL loss (see more details in Appendix A).

### 3.2 THEORETICAL ANALYSIS

To theoretically analyze the abnormal phenomenon above, we conduct gradient analyses by calculating the gradients of both the CE loss and the BinaryKL loss *w.r.t.* the linear classifier and the feature. Our gradient analyses reveal that the gradient of the BinaryKL loss *w.r.t.* the linear classifier

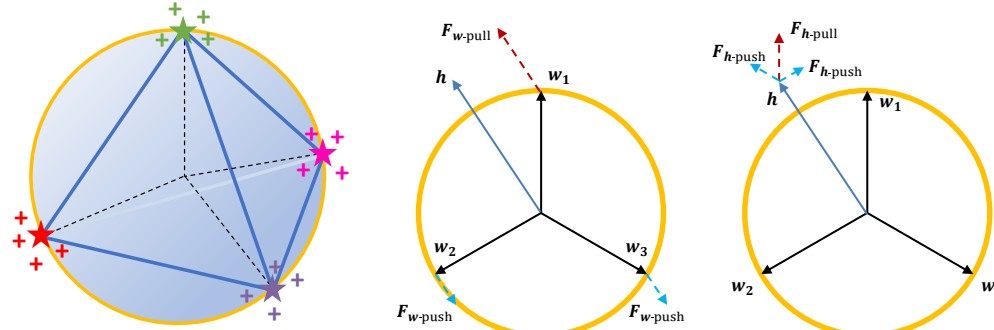

(a) A simplex equiangular tight frame    (b) Gradient directions *w.r.t.* $\boldsymbol{w}$    (c) Gradient directions *w.r.t.* $\boldsymbol{h}$

Figure 3: **Gradient analysis based on the *neural collapse* theory.** (a) An illustration of a simplex ETF when $d = 3$ and $K = 4$. The "+" and "☆" with different colors refer to features and classifier vectors of different classes, respectively. (b) Gradient directions about a certain $\boldsymbol{h}$ (belongs to the 1-st class) *w.r.t.* all $\boldsymbol{w}_i, i \in 1, 2, 3$. (c) Gradient directions *w.r.t.* an $\boldsymbol{h}$ (belongs to the 1-st class).

obstructs it from achieving the ideal state posited by the *neural collapse* theory. Therefore, before delving into the gradient analyses, we provide a brief introduction to the *neural collapse* theory below.

Papyan et al. (2020) revealed the neural collapse phenomenon, where the last-layer features converge to their within-class means, and the within-class means together with the classifier vectors collapse to the vertices of a simplex equiangular tight frame (ETF) at the terminal phase of training. According to Papyan et al. (2020), all vectors in a simplex ETF have an equal $\ell_2$ norm and the same pairwise angle. An illustration of a simplex ETF is shown in Figure 3. The neural collapse phenomenon (Papyan et al., 2020) can be characterized by four manifestations: (1) the variability of the last-layer features in the same class collapses to zero; (2) the class means of different classes' last-layer features converge to a simplex ETF; (3) the linear classifier vectors collapse to the class means; (4) the task of predicting for classification can be simplified into finding out the nearest class center of the last-layer feature. The detailed definition of a simplex ETF and the features of the neural collapse phenomenon can be found in Appendix B.1.

From the above analysis, we can find that a well-trained linear classifier would collapse into a simplex ETF, which implies that any factor that hinders the linear classifier from forming a simplex ETF can adversely affect the training of the linear classifier and thus detrimentally impact the overall training of the student model. It is noteworthy that although the last-layer features in the same class tend to collapse into their mean value as well, this is not the ideal status of a well-trained linear classifier, and such a phenomenon is mainly called over-fitting. Inspired by Yang et al. (2022), we decompose the gradients of $\mathcal{L}_{\mathrm{CE}}$ and $\mathcal{L}_{\mathrm{BinaryKL}}$ to figure out their effects on the linear classifier and the feature extracted by the backbone. First, we decompose the gradients *w.r.t.* the linear classifier. Let $\boldsymbol{h}_{k,i}$ be the feature of the $i$-th item in the $k$-th class extracted by the backbone and $\boldsymbol{w}$ be the linear classifier where $\boldsymbol{w} = [\boldsymbol{w}_1, \ldots, \boldsymbol{w}_K] \in \mathbb{R}^{d \times K}$. Let $n_k$ be the number of instances in the $k$-th class in a certain batch. We denote the $k$-th class output through the softmax function as $p_k(\cdot)$ and the $k$-th class output through the sigmoid function as $q_k(\cdot)$. The detailed definitions of $p_k(\cdot)$ and $q_k(\cdot)$ can be found in Appendix B.2.1.

**Proposition 1** *The gradient of $\mathcal{L}_{\mathrm{overall}}$ w.r.t. the linear classifier can be formulated as follows:*

$$\frac{\partial \mathcal{L}_{\mathrm{overall}}}{\partial \boldsymbol{w}_k} = -(\boldsymbol{F}_{\boldsymbol{w}\text{-pull}}^{\mathrm{CE}} + \alpha \boldsymbol{F}_{\boldsymbol{w}\text{-pull}}^{\mathrm{BinaryKL}}) - (\boldsymbol{F}_{\boldsymbol{w}\text{-push}}^{\mathrm{CE}} + \alpha \boldsymbol{F}_{\boldsymbol{w}\text{-push}}^{\mathrm{BinaryKL}}), \quad (3)$$

*where*

$$\boldsymbol{F}_{\boldsymbol{w}\text{-pull}}^{\mathrm{CE}} = \sum_{i=1}^{n_k}(1 - p_k(\boldsymbol{h}_{k,i}^{\mathcal{S}}))\boldsymbol{h}_{k,i}^{\mathcal{S}}, \quad \boldsymbol{F}_{\boldsymbol{w}\text{-pull}}^{\mathrm{BinaryKL}} = \tau \sum_{i=1}^{n_k}(q_k(\boldsymbol{h}_{k,i}^{\mathcal{T}}) - q_k(\boldsymbol{h}_{k,i}^{\mathcal{S}}))\boldsymbol{h}_{k,i}^{\mathcal{S}},$$

$$\boldsymbol{F}_{\boldsymbol{w}\text{-push}}^{\mathrm{CE}} = -\sum_{k' \neq k}^{K}\sum_{j=1}^{n_{k'}} p_k(\boldsymbol{h}_{k',i}^{\mathcal{S}})\boldsymbol{h}_{k',j}^{\mathcal{S}}, \quad \boldsymbol{F}_{\boldsymbol{w}\text{-push}}^{\mathrm{BinaryKL}} = -\tau \sum_{k' \neq k}^{K}\sum_{j=1}^{n_{k'}}(q_k(\boldsymbol{h}_{k,i}^{\mathcal{S}}) - q_k(\boldsymbol{h}_{k,i}^{\mathcal{T}}))\boldsymbol{h}_{k',j}^{\mathcal{S}}.$$

The proof of Proposition 1 is provided in Appendix B.2.1. The gradient *w.r.t.* $\boldsymbol{w}_k$ can be decomposed into four terms. As Figure 3(b) shows, on one hand, the "pull" terms $\boldsymbol{F}_{\boldsymbol{w}\text{-pull}}^{\text{CE}}$ and $\boldsymbol{F}_{\boldsymbol{w}\text{-pull}}^{\text{BinaryKL}}$ pull $\boldsymbol{w}_k$ towards feature directions of the same class, *i.e.*, $\boldsymbol{h}_{k,i}$; on the other hand, the "push" terms $\boldsymbol{F}_{\boldsymbol{w}\text{-push}}^{\text{CE}}$ and $\boldsymbol{F}_{\boldsymbol{w}\text{-push}}^{\text{BinaryKL}}$ push $\boldsymbol{w}_k$ away from the feature directions of the other classes, *i.e.*, $\boldsymbol{h}_{k',i}$, for all $k' \neq k$. The terms *w.r.t.* the CE loss keep the signs unchanged, which means each coefficient of $\boldsymbol{h}_{k,i}$ in $\boldsymbol{F}_{\boldsymbol{w}\text{-pull}}^{\text{CE}}$ is positive and each coefficient of $\boldsymbol{h}_{k',j}$ in $\boldsymbol{F}_{\boldsymbol{w}\text{-push}}^{\text{CE}}$ is negative.

However, this does not hold for the terms *w.r.t.* the BinaryKL loss. The coefficients of $\boldsymbol{h}_{k,i}$ in $\boldsymbol{F}_{\boldsymbol{w}\text{-pull}}^{\text{BinaryKL}}$ may be negative and the coefficients of $\boldsymbol{h}_{k',j}$ in $\boldsymbol{F}_{\boldsymbol{w}\text{-push}}^{\text{BinaryKL}}$ may be positive. Such coefficients can reduce the absolute values of the gradients in the optimization direction. In the worst-case scenario, suppose the absolute values of these coefficients are larger than those of the CE loss coefficients but have the opposite signs, these coefficients can even guide the optimization to the opposite direction. Hence, the BinaryKL loss may obstruct the linear classifier's learning process. Because a well-trained linear classifier will become a simplex ETF, the terms *w.r.t.* the BinaryKL loss will harm the training of the linear classifier when their corresponding coefficients have the contrary sign to the terms *w.r.t.* the CE loss.

Then, we decompose the gradients *w.r.t.* the feature extracted by the backbone. Suppose the instance $\boldsymbol{x}$ has a feature $\boldsymbol{h}$ and belongs to the $c$-th class. Similar to the gradients *w.r.t.* the linear classifier, we can calculate the gradients *w.r.t.* the features.

**Proposition 2** *The gradient of $\mathcal{L}_{\text{overall}}$ w.r.t. the features can be formulated as follows:*

$$\frac{\partial \mathcal{L}_{\text{overall}}}{\partial \boldsymbol{h}} = -(\boldsymbol{F}_{\boldsymbol{h}\text{-pull}}^{\text{CE}} + \alpha \boldsymbol{F}_{\boldsymbol{h}\text{-pull}}^{\text{BinaryKL}}) - (\boldsymbol{F}_{\boldsymbol{h}\text{-push}}^{\text{CE}} + \alpha \boldsymbol{F}_{\boldsymbol{h}\text{-push}}^{\text{BinaryKL}}), \quad (4)$$

*where*

$$\boldsymbol{F}_{\boldsymbol{h}\text{-pull}}^{\text{CE}} = (1 - p_c(\boldsymbol{h}^{\mathcal{S}}))\boldsymbol{w}_c^{\mathcal{S}}, \quad \boldsymbol{F}_{\boldsymbol{h}\text{-pull}}^{\text{BinaryKL}} = \tau(q_c(\boldsymbol{h}^{\mathcal{T}}) - q_c(\boldsymbol{h}^{\mathcal{S}}))\boldsymbol{w}_c^{\mathcal{S}},$$

$$\boldsymbol{F}_{\boldsymbol{h}\text{-push}}^{\text{CE}} = -\sum_{k \neq c}^{K} p_k(\boldsymbol{h}^{\mathcal{S}})\boldsymbol{w}_k^{\mathcal{S}}, \quad \boldsymbol{F}_{\boldsymbol{h}\text{-push}}^{\text{BinaryKL}} = -\tau \sum_{k \neq c}^{K} (q_k(\boldsymbol{h}^{\mathcal{S}}) - q_k(\boldsymbol{h}^{\mathcal{T}}))\boldsymbol{w}_k^{\mathcal{S}}.$$

The proof of Proposition 2 is provided in Appendix B.2.2. The gradients *w.r.t.* $\boldsymbol{h}$ are decomposed into four terms. As Figure 3(c) shows, the "pull" terms $\boldsymbol{F}_{\boldsymbol{h}\text{-pull}}^{\text{CE}}$ and $\boldsymbol{F}_{\boldsymbol{h}\text{-pull}}^{\text{BinaryKL}}$ pull $\boldsymbol{h}$ towards the directions of the corresponding class vector, *i.e.*, $\boldsymbol{w}_c$, while the "push" terms $\boldsymbol{F}_{\boldsymbol{h}\text{-push}}^{\text{CE}}$ and $\boldsymbol{F}_{\boldsymbol{h}\text{-push}}^{\text{BinaryKL}}$ push $\boldsymbol{h}$ away from the directions of the other class vectors, *i.e.*, $\boldsymbol{w}_k$, for all $k \neq c$. Unlike the linear classifier, a well-trained backbone will not let the features of the same class collapse into a specific vector. The differences among the features of the same class contain essential information from the teacher model. Thus, the terms *w.r.t.* the BinaryKL loss will provide more detailed information about the knowledge learned by the teacher model. As a result, the backbone of the student model can benefit from the terms *w.r.t.* the BinaryKL loss.

### 3.3 OUR PROPOSED METHOD

As demonstrated in our theoretical analysis, the BinaryKL loss facilitates the backbone in learning a better backbone but impedes the linear classifier from achieving an Equiangular Tight Frame status. To fully leverage the positive effects of the BinaryKL loss while mitigating its negative impact, we propose a novel method called Dual-Head Knowledge Distillation (DHKD), which restricts the effects of the BinaryKL loss to the backbone, excluding the original linear classifier. For a given input $\boldsymbol{x}$, we can get a feature $\boldsymbol{h} = f(\boldsymbol{x})$ through the backbone $f(\cdot)$. Just as Figure 4 shows, we use two classifiers for different losses: $g(\cdot)$ for the CE loss and $g'(\cdot)$ for the BinaryKL loss.

The original BinaryKL loss suffers from a problem: when the student output falls into a small neighborhood of the teacher output (which means it is close to the optimization goal), the derivative of the BinaryKL loss at the student output also depends on the value of the teacher output. As a result, the optimization progress cannot be synchronized among distinct teacher outputs with the same $\tau$. Although using different $\tau$ in different settings can lead to state-of-the-art performance, it would introduce more hyper-parameters. To mitigate this issue, we use a variant of the BinaryKL

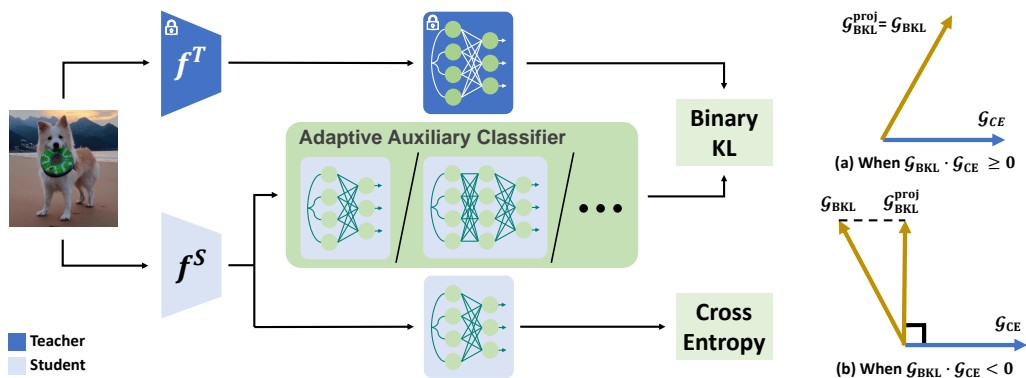

Figure 4: **An illustration of Dual-Head Knowledge Distillation (DHKD).** DHKD decouples the original linear classifier into a duo. We can evade conflicts between two losses by introducing an Adaptive Auxiliary Classifier customized according to the models' architectures. The right side shows the gradient alignment method on CIFAR-100. When the angle between two gradients is larger than $90°$, we will project the gradient of the BinaryKL loss to the orthogonal direction of the gradient of the CE loss.

loss. To unify the gradients at the outputs of the teacher model, we choose to narrow the distance between zero and the difference between the teacher and student. The modified loss (we call it *BinaryKL-Norm*) can be formulated as follows:

$$\mathcal{L}_{\text{BinaryKL-Norm}} = \tau^2 \sum_{i=1}^{B} \sum_{k=1}^{K} \mathcal{KL}\left( \left[ \frac{1}{2}, \frac{1}{2} \right] \middle\| \left[ \sigma\left( \frac{z_{i,k}^{\mathcal{S}} - z_{i,k}^{\mathcal{T}}}{\tau} \right), 1 - \sigma\left( \frac{z_{i,k}^{\mathcal{S}} - z_{i,k}^{\mathcal{T}}}{\tau} \right) \right] \right). \quad (5)$$

It is worth noting that the modification of the loss function does not affect the correctness of our theoretical analysis. The theoretical analysis of the modified loss can be found in Appendix B.3. The performance comparison of the original loss and the modified one is included in Appendix D. Given the challenges of aligning logits between models with different architectures, we relax the constraint on the auxiliary classifier $g'(\cdot)$. Instead of utilizing a linear layer, we employ a one-hidden-layer neural network, which provides additional flexibility and capacity for the model to align the logits effectively when the student model's architecture diverges from the teacher's.

To further improve the stability of the training phase for the CIFAR-100 dataset, we introduce the gradient alignment technique (Yu et al., 2020), which was widely used to handle the conflicts among diverse targets (Gupta et al., 2020; Rame et al., 2022; Zhu et al., 2023). We denote $\frac{\partial \mathcal{L}_{\text{BinaryKL-Norm}}}{\partial \boldsymbol{w}_k}$ as $\mathcal{G}_{\text{BKL}}$ and $\frac{\partial \mathcal{L}_{\text{CE}}}{\partial \boldsymbol{w}_k}$ as $\mathcal{G}_{\text{CE}}$, respectively. The relations between $\mathcal{G}_{\text{BKL}}$ and $\mathcal{G}_{\text{CE}}$ are two-fold. (1) Their angle is smaller than $90°$, which indicates that the optimization direction of the BinaryKL loss does not conflict with the CE loss. In this case, we simply set the updated gradient direction of the auxiliary classifier as $\mathcal{G}_{\text{BKL}}$; (2) Their angle is larger than $90°$, indicating that the BinaryKL loss conflicts with the CE loss. In other words, optimizing the neural network following the BinaryKL loss will weaken the performance of classification. In this case, we project the $\mathcal{G}_{\text{BKL}}$ to the orthogonal direction of $\mathcal{G}_{\text{CE}}$ to optimize the model by the BinaryKL-Norm loss, which avoids increasing the loss for classification. The modified gradient is mathematically formulated as follows:

$$\mathcal{G}_{\text{BKL}}^{\text{proj}} = \begin{cases} \mathcal{G}_{\text{BKL}}, & \text{if } \mathcal{G}_{\text{BKL}} \cdot \mathcal{G}_{\text{CE}} \geq 0, \\ \mathcal{G}_{\text{BKL}} - \frac{\mathcal{G}_{\text{BKL}} \cdot \mathcal{G}_{\text{CE}}}{\|\mathcal{G}_{\text{CE}}\|^2} \mathcal{G}_{\text{CE}}, & \text{otherwise.} \end{cases} \quad (6)$$

By decoupling the linear classifier and doing gradient alignment, we can preserve the positive effects of the BinaryKL-Norm loss on the backbone and simultaneously avoid its negative impacts on the classifier head. The classifier head is only trained with the CE loss without being aligned with the teacher model because we want to induce it into an ideal simplex ETF.

Table 1: **Results on the CIFAR-100 validation.** Teachers and students are in the **same** architectures. The best performance is highlighted in bold, and the second best performance is highlighted with an underline.

| | Teacher | resnet56 72.34 | resnet110 74.31 | resnet32×4 79.42 | WRN-40-2 75.61 | WRN-40-2 75.61 | VGG13 74.64 |
|---|---|---|---|---|---|---|---|
| | Student | resnet20 69.06 | resnet32 71.14 | resnet8×4 72.50 | WRN-16-2 73.26 | WRN-40-1 71.98 | VGG8 70.36 |
| features | FitNet | 69.21 | 71.06 | 73.50 | 73.58 | 72.24 | 71.02 |
| | RKD | 69.61 | 71.82 | 71.90 | 73.35 | 72.22 | 71.48 |
| | CRD | 71.16 | 73.48 | 75.51 | 75.48 | 74.14 | 73.94 |
| | OFD | 70.98 | 73.23 | 74.95 | 75.24 | 74.33 | 73.95 |
| | ReviewKD | 71.89 | 73.89 | 75.63 | 76.12 | 75.09 | **74.84** |
| logits | KD | 70.66 | 73.08 | 73.33 | 74.92 | 73.54 | 72.98 |
| | DKD | **71.97** | **74.11** | 76.32 | 76.24 | 74.81 | 74.68 |
| | DHKD | 71.19 | 73.92 | **76.54** | **76.36** | **75.25** | **74.84** |

Table 2: **Results on the CIFAR-100 validation.** Teachers and students are in **different** architectures. The best performance is highlighted in bold, and the second best performance is highlighted with an underline.

| | Teacher | resnet32×4 79.42 | WRN-40-2 75.61 | VGG13 74.64 | ResNet-50 79.34 | resnet32×4 79.42 |
|---|---|---|---|---|---|---|
| | Student | ShuffleNet-V1 70.50 | ShuffleNet-V1 70.50 | MBN-V2 64.60 | MBN-V2 64.60 | ShuffleNet-V2 71.82 |
| features | FitNet | 73.59 | 73.73 | 64.14 | 63.16 | 73.54 |
| | RKD | 72.28 | 72.21 | 64.52 | 64.43 | 73.21 |
| | CRD | 75.11 | 76.05 | 69.73 | 69.11 | 75.65 |
| | OFD | 75.98 | 75.85 | 69.48 | 69.04 | 76.82 |
| | ReviewKD | **77.45** | 77.14 | **70.37** | 69.89 | 77.78 |
| logits | KD | 74.07 | 74.83 | 67.37 | 67.35 | 74.45 |
| | DKD | 76.45 | 76.70 | 69.71 | 70.35 | 77.07 |
| | DHKD | 76.78 | **77.25** | 70.09 | **71.08** | **77.99** |

# 4 EXPERIMENTS

Information about the comparing methods and implementation details can be found in Appendices E and F. More experimental results and visualizations can be found in Appendices A, G, H, I, J and K.

## 4.1 MAIN RESULTS

**CIFAR-100 image classification.** The validation accuracy on CIFAR-100 is reported in Table 1 and Table 2. Table 1 shows the results where teachers and students are of the same network architectures. Table 2 contains the results where teachers and students are from different architectures. It can be observed that our DHKD achieves remarkable improvements over the vanilla KD on all teacher-student pairs. Compared with the state-of-the-art logit-based method, our method can achieve better performance except in two experiments across ResNet architectures. Furthermore, DHKD achieves comparable or even better performance than feature-based methods, significantly improving the trade-off between performance and training efficiency.

**ImageNet image classification.** The top-1 and top-5 validation accuracy on ImageNet is reported in Table 3 and Table 4. Table 3 shows the results where teachers and students have the same network architectures. Table 4 shows the results where teachers and students have different architectures. We can find that our DHKD achieves comparable or even better performance than the existing methods. The success on the large-scale dataset further proves the effectiveness of our method.

Table 3: **Top-1 and top-5 accuracy (%) on the ImageNet validation.** We set **ResNet-34** as the teacher and **ResNet-18** as the student.

| distillation manner | | | features | | | | logits | | | |
|---|---|---|---|---|---|---|---|---|---|---|
| Metric | Teacher | Student | AT | OFD | CRD | ReviewKD | KD | DKD | DIST | DHKD |
| top-1 | 73.31 | 69.75 | 70.69 | 70.81 | 71.17 | 71.61 | 70.66 | 71.70 | 72.07 | **72.15** |
| top-5 | 91.42 | 89.07 | 90.01 | 89.98 | 90.13 | 90.51 | 89.88 | 90.41 | 90.42 | **90.89** |

Table 4: **Top-1 and top-5 accuracy (%) on the ImageNet validation.** We set **ResNet-50** as the teacher and **MobileNet** as the student.

| distillation manner | | | features | | | | logits | | | |
|---|---|---|---|---|---|---|---|---|---|---|
| Metric | Teacher | Student | AT | OFD | CRD | ReviewKD | KD | DKD | DIST | DHKD |
| top-1 | 76.16 | 68.87 | 69.56 | 71.25 | 71.37 | 72.56 | 68.58 | 72.05 | **73.24** | 72.99 |
| top-5 | 92.86 | 88.76 | 89.33 | 90.34 | 90.41 | 91.00 | 88.98 | 91.05 | 91.12 | **91.45** |

Table 5: **Ablation studies on CIFAR-100.** We choose one pair of models with the same architecture and another pair with different architectures. For the former pair, we set **resnet32×4** as the teacher and **resnet8×4** as the student; for the latter pair, we set **ResNet-50** as the teacher and **MobileNet-V2** as the student.

| CE | BinaryKL -*Norm* | Dual Head | Gradient Alignment | Nonlinear Auxiliary Head | resnet32×4 resnet8×4 | ResNet-50 MBN-V2 |
|---|---|---|---|---|---|---|
| ✓ | | | | | 72.50 | 64.60 |
| | ✓ | | | | 75.15 | 68.52 |
| ✓ | ✓ | | | | N.A. | N.A. |
| ✓ | ✓ | ✓ | | | 76.16 | 70.16 |
| ✓ | ✓ | ✓ | ✓ | | **76.54** | 69.97 |
| ✓ | ✓ | ✓ | ✓ | ✓ | 74.38 | **71.08** |

## 4.2 ABLATION STUDIES

To further analyze how our proposed method improves distillation performance, Table 5 reports the results of the ablation studies on CIFAR-100. We choose one pair of models with the same architecture and another pair with different architectures. It can be observed that most of the performance improvement comes from using our DHKD. Introducing a nonlinear auxiliary classifier helps the pair with different architectures achieve better performance, but it harms the performance of the pair with the same architecture, which confirms the rationality of our selective use of a nonlinear auxiliary classifier. By incorporating these components together, the fusing method achieves the best performance and significantly outperforms the other methods. These results demonstrate that all components are of great importance to the performance of our proposed DHKD.

## 5 CONCLUSION

This paper studied the problem of knowledge distillation. We provided the attempt to combine the BinaryKL loss with the CE loss, while our empirical findings indicated that merging the two losses results in degraded performance, even falling below the performance of using either loss independently. Inspired by previous research on neural collapse, we theoretically demonstrated that while the BinaryKL loss improves the efficacy of the backbone, it conflicts with the CE loss at the level of the linear classifier, thereby leading the model to a sub-optimal situation. To address this issue, we proposed a novel method called Dual-Head Knowledge Distillation, which separates the linear classifier into two distinct parts, each responsible for a specific loss. Experimental results on benchmark datasets confirmed the effectiveness of our proposed method.

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

CONTENTS

## A  Dual Head is the Only Way

As mentioned in Section 3.1, we have tried many other methods before introducing the auxiliary head, including decreasing the balancing parameter $\alpha$, reducing the learning rate, and using a gradient clipping method.

We cannot even give a table for decreasing the balancing parameter $\alpha$ because the training will collapse unless $\alpha$ is less than 0.1, leading to a result even worse than the vanilla KD.

More results about reducing the learning rate and using a gradient clipping method can be found in Table 6 and Table 7. Reducing the learning rate from 0.1 to $1e-2, 5e-3, 1e-3$, the student model still collapses during the training phase most of the time. The only student model that does not collapse has an accuracy of only 39.56%, far below the normally-trained model.

As for the gradient clipping method, even though we change the Maximum Gradient Norm Value (MGNV) in a small step size, the model still cannot achieve comparable performance with the model trained with only the CE loss.

Table 6: **Some attempts that change the learning rate.** Top-1 accuracy on CIFAR-100 is given in the table. For elements that are "N.A.", we give the ordinal number of the epoch when the models collapse during the training phase. It can be observed that almost all the student models collapse. As the learning rate decreases, the time for model collapse is delayed.

| | Same Architecture | | | Different Architectures | | |
|---|---|---|---|---|---|---|
| teacher | resnet56 | resnet110 | resnet32×4 | resnet32×4 | VGG13 | ResNet-50 |
| student | resnet20 | resnet32 | resnet8×4 | ShuffleNet-V1 | MBN-V2 | MBN-V2 |
| lr=1e-2 | N.A.(8) | N.A.(6) | N.A.(5) | N.A.(8) | N.A.(44) | N.A.(34) |
| lr=5e-3 | N.A.(8) | N.A.(8) | N.A.(11) | N.A.(16) | N.A.(48) | N.A.(69) |
| lr=1e-3 | N.A.(35) | N.A.(29) | N.A.(27) | N.A.(50) | N.A.(126) | 39.56 |

Table 7: **Some attempts using gradient clipping (GC) method.** We change the Maximum Gradient Norm Value (MGNV) in a small step size, and the student model still cannot achieve comparable performance with the model trained with only the CE loss. We set **resnet56** as the teacher and **resnet20** as the student. All the experiments are done on CIFAR-100.

| MGNV | 0.01 | 0.1 | 0.3 | 0.5 | 0.7 | 1.0 | Pure CE No GC |
|---|---|---|---|---|---|---|---|
| Test Accuracy | 1.95 | 23.63 | 59.11 | 65.16 | N.A. | N.A. | 69.06 |

## B  Supplementary Material about Neural Collapse

### B.1  Detailed Definition of ETF and Neural Collapse

**Definition 1 (Simplex Equiangular Tight Frame (Papyan et al., 2020))** *A collection of vectors* $\mathbf{m}_i \in \mathbb{R}^d$, $i = 1, 2, \cdots, K$, $d \geq K - 1$, *is said to be a simplex equiangular tight frame if:*

$$\mathbf{M} = \sqrt{\frac{K}{K-1}} \mathbf{U} \left( \mathbf{I}_K - \frac{1}{K} \mathbf{1}_K \mathbf{1}_K^\top \right), \tag{7}$$

*where* $\mathbf{M} = [\mathbf{m}_1, \cdots, \mathbf{m}_K] \in \mathbb{R}^{d \times K}$, $\mathbf{U} \in \mathbb{R}^{d \times K}$ *allows a rotation and satisfies* $\mathbf{U}^\top \mathbf{U} = \mathbf{I}_K$, $\mathbf{I}_K$ *is the identity matrix, and* $\mathbf{1}_K$ *is an all-ones vector.*

**Theorem 1 (Papyan et al. (2020))** *All vectors in a simplex Equiangular Tight Frame (ETF) have an equal* $\ell_2$ *norm and the same pair-wise angle,* i.e.,

$$\mathbf{m}_i^\top \mathbf{m}_j = \frac{K}{K-1} \delta_{i,j} - \frac{1}{K-1}, \forall i, j \in [1, K], \tag{8}$$

*where* $\delta_{i,j}$ *equals to 1 when* $i = j$ *and 0 otherwise. The pairwise angle* $-\frac{1}{K-1}$ *is the maximal equiangular separation of the* $K$ *vectors in* $\mathbb{R}^d$.

Then, according to (Papyan et al., 2020), the neural collapse (NC) phenomenon can be formally described as:

**(NC1)** Within-class variability of the last-layer features collapse: $\Sigma_W \to \mathbf{0}$, and $\Sigma_W :=$ $\mathrm{Avg}_{i,k}\{(\boldsymbol{h}_{k,i} - \boldsymbol{h}_k)(\boldsymbol{h}_{k,i} - \boldsymbol{h}_k)^\top\}$, where $\boldsymbol{h}_{k,i}$ is the last-layer feature of the $i$-th sample in the $k$-th class, and $\boldsymbol{h}_k = \mathrm{Avg}_i\{\boldsymbol{h}_{k,i}\}$ is the within-class mean of the last-layer features in the $k$-th class;

**(NC2)** Convergence to a simplex ETF: $\tilde{\boldsymbol{h}}_k = (\boldsymbol{h}_k - \boldsymbol{h}_G)/\|\boldsymbol{h}_k - \boldsymbol{h}_G\|, k \in [1, K]$, satisfies Eq. (8), where $\boldsymbol{h}_G$ is the global mean of the last-layer features, *i.e.,* $\boldsymbol{h}_G = \mathrm{Avg}_{i,k}\{\boldsymbol{h}_{k,i}\}$;

**(NC3)** Convergence to self duality: $\tilde{\boldsymbol{h}}_k = \boldsymbol{w}_k/\|\boldsymbol{w}_k\|$, where $\boldsymbol{w}_k$ is the classifier vector of the $k$-th class;

**(NC4)** Simplification to the nearest class center prediction: $\arg\max_k \langle \boldsymbol{h}, \boldsymbol{w}_k \rangle = \arg\min_k \|\boldsymbol{h} - \boldsymbol{h}_k\|$, where $\boldsymbol{h}$ is the last-layer feature of a sample to predict for classification.

## B.2 PROOF OF PROPOSITIONS

### B.2.1 PROOF OF PROPOSITION 1

According to (Yang et al., 2022), the negative gradients of $\mathcal{L}_{CE}$ on this batch can be calculated as follows:

$$-\frac{\partial \mathcal{L}_{CE}}{\partial \boldsymbol{w}_k} = \underbrace{\sum_{i=1}^{n_k} \left(1 - p_k\left(\boldsymbol{h}_{k,i}^{\mathcal{S}}\right)\right) \boldsymbol{h}_{k,i}^{\mathcal{S}}}_{\boldsymbol{F}_{\boldsymbol{w}\text{-pull}}^{\mathrm{CE}}} + \underbrace{\left(-\sum_{k'\neq k}^{K} \sum_{j=1}^{n_{k'}} p_k\left(\boldsymbol{h}_{k',j}^{\mathcal{S}}\right) \boldsymbol{h}_{k',j}^{\mathcal{S}}\right)}_{\boldsymbol{F}_{\boldsymbol{w}\text{-push}}^{\mathrm{CE}}}, \tag{9}$$

where $p_k(\boldsymbol{h})$ is the predicted probability that $\boldsymbol{h}$ belongs to the $k$-th class. It is calculated by the softmax function and takes the following form in the CE loss:

$$p_k(\boldsymbol{h}) = \frac{\exp(\boldsymbol{h}^\top \boldsymbol{w}_k)}{\sum_{k'=1}^{K} \exp(\boldsymbol{h}^\top \boldsymbol{w}_{k'})}, \ 1 \leq k \leq K. \tag{10}$$

Similarly, we can calculate the negative gradients of $\mathcal{L}_{\mathrm{BinaryKL}}$ as follows:

$$-\frac{\partial \mathcal{L}_{\mathrm{BinaryKL}}}{\partial \boldsymbol{w}_k} = \underbrace{\sum_{i=1}^{n_k} \tau(q_k(\boldsymbol{h}_{k,i}^{\mathcal{T}}) - q_k(\boldsymbol{h}_{k,i}^{\mathcal{S}}))\boldsymbol{h}_{k,i}^{\mathcal{S}}}_{\boldsymbol{F}_{\boldsymbol{w}\text{-pull}}^{\mathrm{BinaryKL}}} + \underbrace{\left(-\sum_{k'\neq k}^{K} \sum_{j=1}^{n_{k'}} \tau(q_k(\boldsymbol{h}_{k',j}^{\mathcal{S}}) - q_k(\boldsymbol{h}_{k',j}^{\mathcal{T}}))\boldsymbol{h}_{k',j}^{\mathcal{S}}\right)}_{\boldsymbol{F}_{\boldsymbol{w}\text{-push}}^{\mathrm{BinaryKL}}},$$
$$\tag{11}$$

where $q_k(\boldsymbol{h})$ is the binary predicted probability that $\boldsymbol{h}$ has a positive label on the $k$-th class. It is calculated by the sigmoid function and takes the following form in the BinaryKL loss:

$$q_k(\boldsymbol{h}) = \frac{1}{1 + e^{-\boldsymbol{h}^\top \boldsymbol{w}_k/\tau}}. \tag{12}$$

Based upon Equation 9, 11, we can prove Proposition 1.

### B.2.2 PROOF OF PROPOSITION 2

The definitions of $p_k(\boldsymbol{h})$ and $q_k(\boldsymbol{h})$ are the same as Equation 10 and 12.

Then the negative gradients of $\mathcal{L}_{CE}$ *w.r.t.* the features on this batch can be calculated as follows:

$$-\frac{\partial \mathcal{L}_{CE}}{\partial \boldsymbol{h}} = \underbrace{(1 - p_c(\boldsymbol{h}^{\mathcal{S}}))\boldsymbol{w}_c^{\mathcal{S}}}_{\boldsymbol{F}_{\boldsymbol{h}\text{-pull}}^{\text{CE}}} + \underbrace{(-\sum_{k \neq c}^{K} p_k(\boldsymbol{h}^{\mathcal{S}})\boldsymbol{w}_k^{\mathcal{S}})}_{\boldsymbol{F}_{\boldsymbol{h}\text{-push}}^{\text{CE}}}. \tag{13}$$

The negative gradients of $\mathcal{L}_{\text{BinaryKL}}$ can be calculated as follows:

$$-\frac{\partial \mathcal{L}_{\text{BinaryKL}}}{\partial \boldsymbol{w}_k} = \underbrace{\tau(q_c(\boldsymbol{h}^{\mathcal{T}}) - q_c(\boldsymbol{h}^{\mathcal{S}}))\boldsymbol{w}_c^{\mathcal{S}}}_{\boldsymbol{F}_{\boldsymbol{h}\text{-pull}}^{\text{BinaryKL}}} + \underbrace{\left(-\tau\sum_{k \neq c}^{K}(p_k(\boldsymbol{h}^{\mathcal{S}}) - p_k(\boldsymbol{h}^{\mathcal{T}}))\boldsymbol{w}_k^{\mathcal{S}}\right)}_{\boldsymbol{F}_{\boldsymbol{h}\text{-push}}^{\text{BinaryKL}}}. \tag{14}$$

Based upon Equation 13, 14, we can prove Proposition 2.

### B.3  Propositions for the Modified Loss

We can rewrite the $\mathcal{L}_{\text{overall}}$ in the following form:

$$\mathcal{L}_{\text{overall}} = \mathcal{L}_{CE} + \alpha \mathcal{L}_{\text{BinaryKL-Norm}}, \tag{15}$$

where

$$\mathcal{L}_{\text{BinaryKL-Norm}} = \tau^2 \sum_{i=1}^{B} \sum_{k=1}^{K} \mathcal{KL}\left( \left[\frac{1}{2}, \frac{1}{2}\right] \Big\| \left[\sigma\left(\frac{z_{i,k}^{\mathcal{S}} - z_{i,k}^{\mathcal{T}}}{\tau}\right), 1 - \sigma\left(\frac{z_{i,k}^{\mathcal{S}} - z_{i,k}^{\mathcal{T}}}{\tau}\right)\right] \right),$$

as defined in Equation 5.

Define an auxiliary function $w(\cdot, \cdot)$ as below:

$$w_k(\boldsymbol{h}_1, \boldsymbol{h}_2) = \frac{1}{1 + e^{-(\boldsymbol{h}_1^{\top}\boldsymbol{w}_k - \boldsymbol{h}_2^{\top}\boldsymbol{w}_k)/\tau}}. \tag{16}$$

Before proving Proposition 3, we need to prove a lemma:

**Lemma 1** *For all $k' \in \{1, \cdots, K\}$ and $j \in \{1, \cdots, n_{k'}\}$,*

$$\left[(\frac{1}{2} - w_k(\boldsymbol{h}_{k',j}^{\mathcal{S}}, \boldsymbol{h}_{k',j}^{\mathcal{T}}))(\boldsymbol{h}_{k',j}^{\mathcal{S}} - \boldsymbol{h}_{k',j}^{\mathcal{T}})\right]^{\top} \boldsymbol{w}_k \leq 0. \tag{17}$$

**Proof.** If $(\boldsymbol{h}_{k',j}^{\mathcal{S}} - \boldsymbol{h}_{k',j}^{\mathcal{T}})^{\top}\boldsymbol{w}_k > 0$, then $w_k(\boldsymbol{h}_{k',j}^{\mathcal{S}}, \boldsymbol{h}_{k',j}^{\mathcal{T}}) > \frac{1}{2}$. If $(\boldsymbol{h}_{k',j}^{\mathcal{S}} - \boldsymbol{h}_{k',j}^{\mathcal{T}})^{\top}\boldsymbol{w}_k \leq 0$, then $w_k(\boldsymbol{h}_{k',j}^{\mathcal{S}}, \boldsymbol{h}_{k',j}^{\mathcal{T}}) \leq \frac{1}{2}$. So, in both conditions, we have the following inequality:

$$\left[(\frac{1}{2} - w_k(\boldsymbol{h}_{k',j}^{\mathcal{S}}, \boldsymbol{h}_{k',j}^{\mathcal{T}}))(\boldsymbol{h}_{k',j}^{\mathcal{S}} - \boldsymbol{h}_{k',j}^{\mathcal{T}})\right]^{\top} \boldsymbol{w}_k = (\frac{1}{2} - w_k(\boldsymbol{h}_{k',j}^{\mathcal{S}}, \boldsymbol{h}_{k',j}^{\mathcal{T}})) \left[(\boldsymbol{h}_{k',j}^{\mathcal{S}} - \boldsymbol{h}_{k',j}^{\mathcal{T}})^{\top}\boldsymbol{w}_k\right] \leq 0.$$

Then, we can get the propositions for the modified loss.

**Proposition 3** *The gradients of $\mathcal{L}_{\text{overall}}$ w.r.t. the linear classifier can be formulated as follows:*

$$\frac{\partial \mathcal{L}_{\text{overall}}}{\partial \boldsymbol{w}_k} = -(\boldsymbol{F}_{\boldsymbol{w}\text{-pull}}^{\text{CE}} + \boldsymbol{F}_{\boldsymbol{w}\text{-push}}^{\text{CE}}) - \alpha \boldsymbol{F}_{\boldsymbol{w}-\text{obstacle}}^{\text{BinaryKL-Norm}}, \tag{18}$$

*where*

$$\boldsymbol{F}_{\boldsymbol{w}\text{-pull}}^{\text{CE}} = \sum_{i=1}^{n_k}(1 - p_k(\boldsymbol{h}_{k,i}^{\mathcal{S}}))\boldsymbol{h}_{k,i}^{\mathcal{S}}, \quad \boldsymbol{F}_{\boldsymbol{w}\text{-push}}^{\text{CE}} = -\sum_{k'\neq k}^{K}\sum_{j=1}^{n_{k'}} p_k(\boldsymbol{h}_{k',i}^{\mathcal{S}})\boldsymbol{h}_{k',j}^{\mathcal{S}},$$

$$\boldsymbol{F}_{\boldsymbol{w}-\text{obstacle}}^{\text{BinaryKL-Norm}} = \tau \sum_{k'=1}^{K}\sum_{j=1}^{n_{k'}}(\frac{1}{2} - w_k(\boldsymbol{h}_{k',j}^{\mathcal{S}}, \boldsymbol{h}_{k',j}^{\mathcal{T}}))(\boldsymbol{h}_{k',j}^{\mathcal{S}} - \boldsymbol{h}_{k',j}^{\mathcal{T}}). \tag{19}$$

**Proof.** The definition of $p_k(\boldsymbol{h})$ is the same as Equation 10. The negative gradients of $\mathcal{L}_{\text{BinaryKL-Norm}}$ on this batch can be calculated as follows:

$$-\frac{\partial \mathcal{L}_{\text{BinaryKL-Norm}}}{\partial \boldsymbol{w}_k} = \sum_{k'=1}^{K}\sum_{j=1}^{n_{k'}}\tau(\frac{1}{2} - w_k(\boldsymbol{h}_{k',j}^{\mathcal{S}}, \boldsymbol{h}_{k',j}^{\mathcal{T}}))(\boldsymbol{h}_{k',j}^{\mathcal{S}} - \boldsymbol{h}_{k',j}^{\mathcal{T}}) \tag{20}$$

According to Lemma 1, each term above has an opposite direction with $\boldsymbol{w}_k$, which would obstruct the learning of the linear classifier. Consequently, we can denote it as $\boldsymbol{F}_{\boldsymbol{w}-\text{obstacle}}^{\text{BinaryKL-Norm}}$.

Based upon Equation 9, 20, we can prove Proposition 3.

**Remark** The form of Proposition 3 differs a lot from Proposition 1, but Proposition 3 shows that the BinaryKL-Norm loss has stronger negative effects over the linear classifier the BinaryKL loss: it hinders the training of the linear classifier all the time without deploy any positive effect.

**Proposition 4** *The gradients of $\mathcal{L}_{\text{overall}}$ w.r.t. the features can be formulated as follows:*

$$\frac{\partial \mathcal{L}_{\text{overall}}}{\partial \boldsymbol{h}} = -(\boldsymbol{F}_{\boldsymbol{h}\text{-pull}}^{\text{CE}} + \alpha \boldsymbol{F}_{\boldsymbol{h}\text{-pull}}^{\text{BinaryKL-Norm}}) - (\boldsymbol{F}_{\boldsymbol{h}\text{-push}}^{\text{CE}} + \alpha \boldsymbol{F}_{\boldsymbol{h}\text{-push}}^{\text{BinaryKL-Norm}}), \tag{21}$$

*where*

$$\boldsymbol{F}_{\boldsymbol{h}\text{-pull}}^{\text{CE}} = (1 - p_c(\boldsymbol{h}^{\mathcal{S}}))\boldsymbol{w}_c^{\mathcal{S}}, \quad \boldsymbol{F}_{\boldsymbol{h}\text{-pull}}^{\text{BinaryKL-Norm}} = \tau(\frac{1}{2} - w_c(\boldsymbol{h}^{\mathcal{S}}, \boldsymbol{h}^{\mathcal{T}}))\boldsymbol{w}_c^{\mathcal{S}},$$

$$\boldsymbol{F}_{\boldsymbol{h}\text{-push}}^{\text{CE}} = -\sum_{k\neq c}^{K} p_k(\boldsymbol{h}^{\mathcal{S}})\boldsymbol{w}_k^{\mathcal{S}}, \quad \boldsymbol{F}_{\boldsymbol{h}\text{-push}}^{\text{BinaryKL-Norm}} = \tau \sum_{k\neq c}^{K}(w_k(\boldsymbol{h}^{\mathcal{S}}, \boldsymbol{h}^{\mathcal{T}}) - \frac{1}{2})\boldsymbol{w}_k^{\mathcal{S}}. \tag{22}$$

**Proof.** The definition of $p_k(\boldsymbol{h})$ is the same as Equation 10. The negative gradients of $\mathcal{L}_{\text{BinaryKL-Norm}}$ can be calculated as follows:

$$-\frac{\partial \mathcal{L}_{\text{BinaryKL-Norm}}}{\partial \boldsymbol{w}_k} = \underbrace{\tau(\frac{1}{2} - w_c(\boldsymbol{h}^{\mathcal{S}}, \boldsymbol{h}^{\mathcal{T}}))\boldsymbol{w}_c^{\mathcal{S}}}_{\boldsymbol{F}_{\boldsymbol{h}\text{-pull}}^{\text{BinaryKL-Norm}}} + \underbrace{\left(-\tau \sum_{k\neq c}^{K}(w_k(\boldsymbol{h}^{\mathcal{S}}, \boldsymbol{h}^{\mathcal{T}}) - \frac{1}{2})\boldsymbol{w}_k^{\mathcal{S}}\right)}_{\boldsymbol{F}_{\boldsymbol{h}\text{-push}}^{\text{BinaryKL-Norm}}}. \tag{23}$$

Based upon Equation 13, 23, we can prove Proposition 4.

## C  LIMITATIONS

The CE loss and the logit-level loss are not universally incompatible. For instance, the MSE loss is not incompatible with the CE loss. So, our solution is not necessary for all of the logit-level losses. It is only a requisite for the BinaryKL loss and the BinaryKL-Norm loss.

Because the *neural collapse* theory mainly focuses on the classification problem, we do not consider the other tasks, such as object detection, and only test our method on the classification task.

# D    COMPARISON BETWEEN BINARYKL AND BINARYKL-NORM

The original BinaryKL loss encounters an issue wherein, as the student output approaches a small neighborhood of the teacher output-—indicative of nearing the optimization target-—-the derivative of the BinaryKL loss at the student output also depends on the value of the teacher output. As a result, the optimization progress cannot be synchronized with distinct teacher outputs, which may lead to sub-optimal performance.

From Table 8 and 9, we can find that using the original BinaryKL loss with a fixed $\tau = 2$ cannot achieve the best performance. With the best $\tau$ for different settings, the original BinaryKL loss can achieve comparable performance with the BinaryKL-Norm loss with a fixed $\tau = 2$. Consequently, we choose to use the BinaryKL-Norm loss to decrease the number of hyper-parameters.

Table 8: **Comparison between BinaryKL and BinaryKL-Norm.** For the BinaryKL loss, we compare the results with a fixed $\tau = 2$ and the best $\tau$. Teachers and students are in the **same** architectures. All the experiments are done on the CIFAR-100 dataset.

| Teacher | resnet56 | resnet110 | resnet32×4 | WRN-40-2 | WRN-40-2 | VGG13 |
|---|---|---|---|---|---|---|
| Student | resnet20 | resnet32 | resnet8×4 | WRN-16-2 | WRN-40-1 | VGG8 |
| DHKD using BinaryKL with $\tau = 2$ | 68.95 | 72.64 | 76.56 | 75.24 | 73.24 | 74.26 |
| DHKD using BinaryKL with the best $\tau$ | 70.23 | 73.11 | 76.86 | 76.18 | 74.15 | 74.78 |
| DHKD using BinaryKL-Norm with $\tau = 2$ | 71.19 | 73.92 | 76.54 | 76.36 | 75.25 | 74.84 |

Table 9: **Comparison between BinaryKL and BinaryKL-Norm.** For the BinaryKL loss, we compare the results with a fixed $\tau = 2$ and the best $\tau$. Teachers and students are in **different** architectures. All the experiments are done on the CIFAR-100 dataset.

| Teacher | resnet32×4 | WRN-40-2 | VGG13 | ResNet-50 | resnet32×4 |
|---|---|---|---|---|---|
| Student | ShuffleNet-V1 | ShuffleNet-V1 | MBN-V2 | MBN-V2 | ShuffleNet-V2 |
| DHKD using BinaryKL with $\tau = 2$ | 75.60 | 75.92 | 69.85 | 69.54 | 77.63 |
| DHKD using BinaryKL with the best $\tau$ | 77.04 | 76.67 | 70.03 | 70.66 | 77.82 |
| DHKD using BinaryKL-Norm with $\tau = 2$ | 76.78 | 77.25 | 70.09 | 71.08 | 77.99 |

# E    COMPARING METHODS.

To validate the proposed method, we compare it with the following sota KD methods:

- FitNet (Romero et al., 2015), the first approach distills knowledge from intermediate features by measuring the distance between feature maps;
- RKD (Park et al., 2019), which captures the relations among instances to guide the training of the student model;
- CRD (Tian et al., 2019), which incorporates contrastive learning into knowledge distillation;
- OFD (Heo et al., 2019), which introduces a new distance function to align features in the same stage between the teacher and student using marginal ReLU;
- ReviewKD (Chen et al., 2021), which transfers knowledge across different stages instead of just focusing on features in the same levels;
- DKD (Zhao et al., 2022), which decouples the vanilla KD method into two parts and assigns different weights to different parts;

- DIST (Huang et al., 2022), which relaxes the exact matching in previous KL divergence loss with a correlation-based loss and performs better when the discrepancy between teacher and student is large.

## F    IMPLEMENTATION DETAILS

We perform experiments on two benchmark datasets: CIFAR-100 (Krizhevsky & Hinton, 2009) and ImageNet (Deng et al., 2009). CIFAR-100 covers 100 categories. It contains 50,000 images in the train set and 10,000 images in the test set. ImageNet covers 1,000 categories of images. It contains 1.28 million images in the train set and 50,000 images in the test set.

**CIFAR-100**: Teachers and students are trained for 240 epochs with SGD, and the batch size is 64. The learning rates are 0.01 for ShuffleNet (Zhang et al., 2018a; Ma et al., 2018) and MobileNet-V2 (Sandler et al., 2018), and 0.05 for the other series (*e.g.* VGG(Simonyan & Zisserman, 2015), ResNet (He et al., 2016) and WRN (Zagoruyko & Komodakis, 2016)). The learning rate is divided by 10 at the 150th, 180th, and 210th epochs. The weight decay and the momentum are set to 5e-4 and 0.9. The weight for the CE loss is set to 1.0, and the temperature is set to 2 for all experiments. $\alpha$ is set as 0.1 for (resnet56→resnet20, resnet110→ resnet32) and is chosen from $\{0.5, 1, 2\}$ for all other experiments. We choose a linear classifier as the auxiliary classifier for students having the same architectures as their teachers. A one-hidden-layer MLP with 200 hidden neurons is deployed as the auxiliary classifier for students with different architectures from their teachers. All the experiments on CIFAR-100 are conducted on GeForce RTX 3090 GPUs.

**ImageNet**: Our implementation for ImageNet follows the standard practice. We train the models for 100 epochs. The batch size is 256, and the learning rate is initialized to 0.1 and divided by 10 for every 30 epochs. Weight decay is 1e-4, and the weight for the CE loss is set to 1.0. We set the temperature as 2 and $\alpha$ as 0.1 for all experiments. We only use the BinaryKL loss in the first 50 epochs of the training phase, which means that we set $\alpha = 0$ for the last 50 epochs. **We do not use the gradient alignment method on ImageNet** because it would seriously slow down the training speed, and we can still achieve the SOTA performance without it. Strictly following (Zhao et al., 2022), for distilling networks of the same architecture, the teacher is ResNet-34 model, the student is ResNet-18; for different series, the teacher is ResNet-50 model, the student is MobileNet-V1. All the experiments on ImageNet are conducted on H100 PCIe GPUs.

## G    T-SNE VISUALIZATION OF THE FEATURES

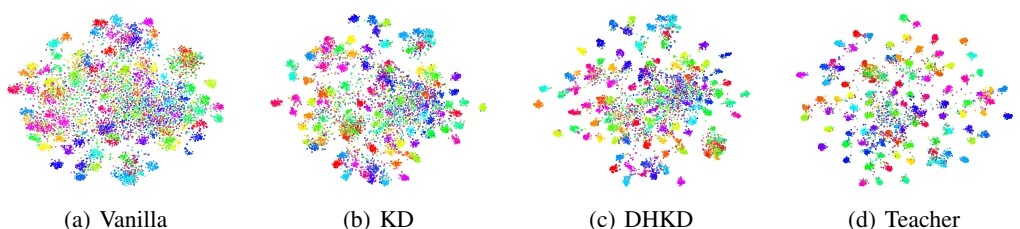

| (a) Vanilla | (b) KD | (c) DHKD | (d) Teacher |

Figure 5: **t-SNE visualization of features learned by different methods.** We do the visualization on the test set of CIFAR-100. We set **resnet32×4** as the teacher and **resnet8×4** as the student.

t-SNE (t-distributed Stochastic Neighbor Embedding) (Van der Maaten & Hinton, 2008) is a famous unsupervised non-linear dimensionality reduction technique for visualizing high-dimensional data. We use t-SNE to visualize the features (the outputs of the models' backbones) learned by different models: vanilla student without distillation, models trained by KD, DHKD, and the teacher model.

The t-SNE results show that the features of KD are more separable than the vanilla student model, and the features of DHKD are more separable than KD but are less than the ones of the teacher model. We attribute this to the restriction of the model capacity. The t-SNE results prove that DHKD improves the discriminability of deep features.

# H  DIFFERENCE OF CORRELATION MATRICES

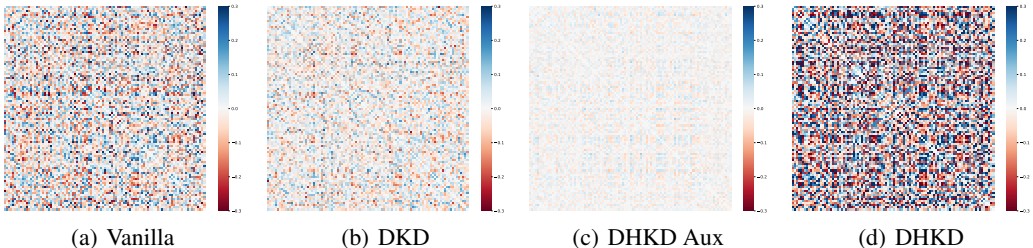

(a) Vanilla      (b) DKD      (c) DHKD Aux      (d) DHKD

Figure 6: **The difference between the correlation matrices of the teacher's and student's logits.** "DHKD Aux" is short for DHKD's auxiliary head. We do the visualization on the test set of CIFAR-100, with **resnet32×4** as the teacher and **resnet8×4** as the student. The performance sorting of four classifier heads is DHKD (76.54) > DKD (76.37) > DHKD's Aux (76.20) > Vanilla (72.17).

We compute the differences between the correlation matrices of the teacher's and student's logits for three different students: the vanilla student without distillation and the students trained by DKD and DHKD. Notably, the student model trained by DHKD has two classifier heads, thus having two sets of logits. We do the visualization for both heads. It can be found that the auxiliary head of DHKD captures the most correlation structure in the logits, as shown by the smallest differences between the teacher and the student. However, the original head of DHKD shows the biggest differences between the teacher and the student, even bigger than the vanilla student model. We attribute this phenomenon to the fact that we only align the output of the auxiliary classifier with the teacher in DHKD and the logits output by the original classifier is never aligned with the teacher.

# I  TRAINING TIME

We evaluate the training costs of state-of-the-art KD methods, demonstrating that our DHKD method also exhibits high training efficiency. As presented in Table 10, our DHKD strikes a good balance between model performance and training costs (*e.g.*, training time and additional parameters). Since DHKD only introduces a classifier head to the original model, it only adds a little computational complexity compared to traditional KD.

We also recorded the training time consumption on ImageNet. As stated in Section F, we utilize the gradient alignment module only to achieve state-of-the-art performance on CIFAR-100 and do not apply it to ImageNet. Consequently, our method only needs 616 ms for each batch, nearly equal to DKD which costs 614 ms.

Table 10: Training time (per batch) and the numbers of extra parameters on CIFAR-100. We set resnet32×4 as the teacher and resnet8×4 as the student.

| Method | KD | RKD | FitNet | OFD | CRD | ReviewKD | DKD | DHKD w/o GA | DHKD w/ GA |
|---|---|---|---|---|---|---|---|---|---|
| Top-1 Accuracy | 73.33 | 71.90 | 73.50 | 74.95 | 75.51 | 75.63 | 76.32 | 76.16 | 76.54 |
| Time (ms) | 11 | 25 | 14 | 19 | 41 | 26 | 11 | 14 | 21 |
| # params | 0 | 0 | 16.8k | 86.9k | 12.3M | 1.8M | 0 | 25.6k | 25.6k |

# J  PARAMETER SENSITIVITY ANALYSIS

We study the influence of hyper-parameters in this section. In all of our experiments, the temperature parameter $\tau$ is fixed at 2. Therefore, the only adjustable hyperparameter is $\alpha$. Table 11 illustrates the performance of DHKD as the value of $\alpha$ changes among 0.2, 0.5, 1, 2, 5 on CIFAR-100. From the table, it can be observed that the performance of DHKD is not very sensitive to the parameter $\alpha$.

Table 11: The parameter sensitivity analysis on CIFAR-100. We set resnet32×4 as the teacher and resnet8×4 as the student.

| $\alpha$ | 0.2 | 0.5 | 1 | 2 | 5 |
|---|---|---|---|---|---|
| Top-1 Accuracy | 75.66 | 75.85 | 76.24 | 76.54 | 67.56 |

## K  THE PERFORMANCE OF "DUAL HEAD + VANILLA KD"

In this section, we conduct the experiments of comparing DHKD and "dual head + vanilla KD". The results are shown in Table 12, illustrating that our dual-head strategy with KL loss on the predicted probability provides only a slight improvement. As a result, only adding a new classifier cannot bring the improvement as much as our method, which does not undermine our contribution.

Table 12: Comparison between DHKD and "dual head + vanilla KD".

| Teacher | Student | without distillation | vanilla KD | dual head + vanilla KD | DHKD |
|---|---|---|---|---|---|
| WRN-40-2 | WRN-16-2 | 73.26 | 74.92 | 75.24 | 76.36 |
| ResNet50 | MBN-V2 | 64.60 | 67.35 | 68.76 | 71.08 |

