# OpenReview forum: "Dual-Head Knowledge Distillation: Enhancing Logits Utilization with an Auxiliary Head"
_ICLR.cc/2025/Conference — ICLR 2025 Conference Withdrawn Submission_

### Official Review · Reviewer_uuFN · 2024-10-20

**Soundness:** 3
**Presentation:** 3
**Contribution:** 3
**Rating:** 6
**Confidence:** 4

**Summary:**

This paper aims to introduce the recently proposed binary KL loss to perform logit distillation. The authors identify the incompatibility between the binary KL loss and the CE loss during distillation. Therefore, an auxiliary classification head is introduced to mitigate this problem. Experiments are performed on various datasets and teacher-student pairs.

**Strengths:**

(1) The paper is well-written and easy to follow.

(2) The motivation for using an auxiliary classification head for distillation is interesting. The authors empirically and theoretically demonstrate the conflict between the CE loss and the Binary KL loss, and therefore, introduce an auxiliary head to decouple these two losses.

(3) The ablation study demonstrates that the introduced auxiliary head significantly improves the distillation performance by using the CE loss and the Binary KL loss.

**Weaknesses:**

(1) The effectiveness of the proposed method is limited as shown in Tables 1-4. The proposed method only achieves 0.1~0.2% improvements in most cases and underperforms other methods in certain cases.

(2) In addition to the marginal improvements, the proposed method requires an additional classification head for distillation, which increases the computation and memory costs of the proposed method. In terms of the results of distillation accuracy and training costs, the proposed method does not perform better than the existing logit distillation methods.

(3) The comparing methods are not new. The latest distillation methods used in this paper, i.e., DKD and DIST were published in 2022. There are several advanced distillation methods (e.g., [1] and [2]) can be added for comparisons.

[1] Logit Standardization in Knowledge Distillation, CVPR 2024

[2] VkD: Improving Knowledge Distillation using Orthogonal Projections, CVPR 2024

**Questions:**

(1) My major concern lies in the effectiveness of the proposed method as stated in the Weaknesses section. The training complexity of the proposed method is higher while the improvements are limited. Therefore, I suggest that the authors explore more experimental settings (e.g., transfer learning) to illustrate the advantages of the proposed method.

(2) The proposed dual head method is similar to [3], which also adds a projection head to the student logits to assist distillation. Therefore, I am wondering about the performance of adding the auxiliary head after the student logits instead of the student features.

(3) After distillation, which head should be used for inference?

[3] Understanding the Effects of Projectors in Knowledge Distillation, arXiv 2023.

---

### Official Review · Reviewer_qJDY · 2024-10-24

**Soundness:** 1
**Presentation:** 4
**Contribution:** 2
**Rating:** 5
**Confidence:** 3

**Summary:**

This paper introduces the logit-level BinaryKL loss to distill the inherent knowledge of logit vectors—which other knowledge distillation (KD) methods overlook due to the softmax function—from teacher models to student models. However, simply combining the BinaryKL loss with the cross-entropy (CE) loss results in conflicting gradients exclusively within the linear classifier of the student model during training, while the backbone remains unaffected. This paper provide a theoretical explanation for this issue, leveraging the concept of a simplex equiangular tight frame from neural collapse theory.  To address this problem, DHKD proposes adding an auxiliary classifier head, which is used solely to propagate gradients to the backbone for BinaryKL loss. This auxiliary head is decoupled from the original classifier head responsible for CE loss. However, the authors observed that BinaryKL loss suffers from unsynchronized optimization among the teacher’s outputs with the same temperature parameter ($\tau$). Instead of assigning different $\tau$ values, DHKD introduces BinaryKL-Norm, which mitigates the issue without compromising the theoretical guarantees. Additionally, DHKD adjusts the number of hidden layers in the linear classifier based on the structural differences between the teacher and student models and employs a gradient alignment technique to enhance performance.

**Strengths:**

The analysis of compatibility with distillation loss, framed in terms of gradients based on neural gradient theory, is novel and has the potential to inspire other researchers. Additionally, the paper’s claims are well-justified, and the flow is clear and well-structured.

**Weaknesses:**

The motivation for introducing BinaryKL loss is unclear. While this paper argues that traditional knowledge distillation methods overlook the difference embedded in logit vectors with identical softmax outputs, it fails to explain what distinct information these logit vectors contain despite having the same softmax output, and why this information is important. Specifically, the paper does not clarify what useful information the teacher model transfers through the logit vectors or why this information is beneficial to student. Furthermore, this paper applies the BinaryKL loss, which is similar to the binary cross-entropy loss used in multi-label learning proposed by Yang et al. (2023), and extends it to BinaryKL-Norm loss. However, the paper does not explain the unique advantages of BinaryKL loss compared to other loss methods that directly compare two logit vectors  without converting them into probabilities for each class (e.g., L1 loss, L2 loss, and cosine similarity). Could you clarify why it is necessary to compare distributions for each label between the student and teacher models to effectively transfer logit information?

Proposition 2 and the application of the gradient alignment technique do not seem to sufficiently support the claim that there are no conflicts between the gradients of the CE loss and the BinaryKL loss on the backbone. To demonstrate this, the paper needs to theoretically show in Proposition 2 that the coefficients of $w_c^\mathcal{S}$ are positive in $F_{h-pull}^{BinaryKL}$ and those of $w_k^\mathcal{S}$ are negative in $F_{h-push}^{BinaryKL}$ under all conditions, as well as superior performance without the gradient alignment technique. Additionally, the auxiliary classifier does not appear to be directly related to the main concept of BinaryKL-Norm or the inherent information in the logit vectors.

**Questions:**

1. What distinct information is contained in logit vectors that have the same softmax output but different values, and why is this information from the teacher important and beneficial to the student?
2. What are the unique advantages of BinaryKL loss compared to other loss methods that directly compare two logit vectors without converting them into probabilities for each class (e.g., L1 loss, L2 loss, and cosine similarity)?
3. Could you demonstrate that the coefficients of $w_c^\mathcal{S}$ are positive in $F_{h-pull}^{BinaryKL}$ and those of $w_k^\mathcal{S}$ are negative in $F_{h-push}^{BinaryKL}$ under all conditions?
4. Why is the gradient alignment technique necessary if there are no conflicts between the gradients of the CE loss and the BinaryKL loss on the backbone?
5. How many experiments did you run for each result reported in this paper?
6. could you explain me the meaning of the sentence,  "This is not the ideal status of a well-trained linear classifier, and such a phenomenon is mainy called over-fitting" on page 5?
7. could you explain me the meaning of the sentence,  "the optimization progress cannot be synchronized among distinct teacher outputs with the same $\tau$." on page 6?

---

### Official Review · Reviewer_XzUc · 2024-11-04

**Soundness:** 2
**Presentation:** 2
**Contribution:** 2
**Rating:** 3
**Confidence:** 4

**Summary:**

This paper proposes a new knowledge distillation loss by splitting the linear classifier of the student model into two separate classification heads for the computation of a Binary KL loss and a Cross-Entropy loss. In this way, the student could improve the performance of the student model.

**Strengths:**

This paper proposes a new knowledge distillation loss and empirical experiments verify its effectiveness.

**Weaknesses:**

1. This paper empirically finds that combing the probability-level CE loss and the logit-level BinaryKL loss can degrade the student model’s performance and suggests to use a classifier-splitting strategy to alleviate this issue. However, the necessity of using this BinaryKL loss is unclear. As far as I know, this kind of logit-level loss is neither a state-of-the-art choice nor widely used in practice. In contrast, Mean-Square-Error [1][2] applied on logits or modified logits with the classifier [3][4][5] is a more popular choice for logit-level distillation. Additionally, this paper lacks sufficient discussion on its approach’s differences from these established methods.

2. The “information loss through softmax” illustrated in Fig 1 is a well-known characteristic of the softmax operator, i.e., shifting all logits by a constant does not affect the softmax output . However, this paper omits citations of prior works (e.g., [1][2]) after these statements and lacks comparison with their results.

[1] https://arxiv.org/abs/1312.6184

[2] https://arxiv.org/abs/2105.08919

[3] https://openreview.net/forum?id=ZzwDy_wiWv

[4] https://arxiv.org/abs/2203.14001

[5] https://arxiv.org/abs/2403.01427

**Questions:**

Please refer to the weaknesses above.

---

### Note · Authors · 2024-11-13

I have read and agree with the venue's withdrawal policy on behalf of myself and my co-authors.